# Menthone Inhalation Alleviates Local and Systemic Allergic Inflammation in Ovalbumin-Sensitized and Challenged Asthmatic Mice

**DOI:** 10.3390/ijms23074011

**Published:** 2022-04-04

**Authors:** Yi-Hsuan Su, Jin-Yuarn Lin

**Affiliations:** Department of Food Science and Biotechnology, National Chung Hsing University, Taichung 40227, Taiwan; shuan225@hotmail.com

**Keywords:** *Ccr3* gene expression, eosinophilia, Th1/Th2 immune balance

## Abstract

Menthone is rich in *Mentha* × *Piperita* L. essential oil and it has anti-inflammatory properties; research shows that it is useful, via percutaneous absorption, in treating inflammation-related diseases. However, anti-allergic inflammatory effects of volatile menthone have not yet been used to treat allergic asthma, in vivo. We hypothesized that menthone inhalation may have anti-inflammatory and anti-allergic effects in patients with allergic asthma. Therefore, in our study, menthone inhalation was used to treat ovalbumin (OVA)-sensitized and challenged asthmatic mice. Allergic inflammation mediator changes in the lungs and airways, sera, splenocytes, and peritoneal macrophages of the mice were measured. Relative expression amounts of six receptor genes related to allergic inflammation of the lungs and airways were quantitated using a two-step real time quantitative polymerase chain reaction (qPCR). Results showed that menthone inhalation increased serum OVA-specific IgG2a/IgG1 and IgG2a/IgE ratios, increased Th1-type cytokine production in the bronchoalveolar lavage fluid, and decreased nitric oxide, protein, and eotaxin levels. Menthone inhalation inhibited mast cell and eosinophil degranulation, and *chemokine (C-C motif) receptor 3* (*Ccr3*) gene expression amounts, but (relatively) increased Th1 cytokine secretion by splenocytes. Our results evidence that menthone inhalation alleviates local and systemic allergic inflammation in asthmatic mice.

## 1. Introduction

Asthma, a Th2-skewed common chronic disease, is characterized by the production of large quantities of serum IgE antibodies by B lymphocytes, and decreasing type 1 helper T cell (Th1) (IFN-γ)/Th2 (IL-4) cytokine ratios by CD4^+^ T lymphocytes [1]. Research shows a vital role of CD4^+^ T lymphocytes in allergic asthmatic patients and studies have been conducted on many antigen-challenged murine models [2]. Numerous activated CD4^+^ T lymphocytes were found in the bronchial mucosa of asthmatic patients, while various cytokine levels by CD4^+^ T lymphocytes, including Th1, Th2, Th9, and Th17 cells, augmented antigenic challenges, subsequently accompanied with local eosinophilia and bronchial hyperresponsiveness (BHR) [2]. Changes involving T cell cytokine levels, differential cell counts, inflammatory mediators, etc., in bronchoalveolar lavage fluid (BALF), as well as systemic immune responses in asthmatic subjects, may reflect the status of an asthma attack and the effectiveness of asthma management through either classical or alternative therapies [1].

Mast cells and eosinophils are also involved in allergic diseases, because they can release pre-stored intra-granules or newly synthesized mediators by degranulation or secretion when they are activated by various secretagogues, e.g., small chemical agents and large allergic proteins, such as ovalbumin [3,4]. Mast cells, after activated by secretagogues, may increase intracellular free calcium concentration through a specific calcium channel, granule trafficking, or exocytosis [4]. Moreover, mast cells can be activated through an IgE cross-linkage with its specific allergen on the mast cell surface, immediately resulting in mast cell degranulation, subsequently inducing different allergic symptoms due to the release of different mediators, including histamine, tryptase, and β-hexosaminidase [1,2,4,5]. Steroids, semi-synthetic and synthetic drugs, and natural mast cell stabilizing agents, including terpenoids, phenols, flavonoids, coumarins, amino acids, etc., have been clinically applied to prevent allergic reactions from mast cell degranulation [3].

Currently, many drugs are widely used for asthma therapies, targeting different reactions; however, these drugs, particularly steroids, may have some adverse side effects [6]. Traditional medicine (e.g., natural products) are alternative or complementary treatments for asthma [6]. Mast cell stabilizers from *Phlomis umbrosa* root, eucalyptus oil, 8-hydroxyisocapnolactone-2-3-diol, and friedelin (a triterpenoid ketone) have been used to inhibit allergic response, mast cell degranulation, inflammatory cytokine secretion, and IgE-FcεRI signaling [3,7,8,9]. Recently, particular terpenoids, which are secondary metabolic products rich in different herbal plants and marine organisms, with volatile, antioxidant, anti-inflammatory, and/or Th1/Th2 immunomodulatory characteristics, have been noticed due to their pharmacological potential in the treatment of allergic asthma [1,10,11,12,13,14,15]. Among numerous terpenoids, monoterpenes have demonstrated pharmacological potential in the treatment of inflammation [16]. Recently, menthone, a volatile mono-terpenoid compound with strong immunomodulatory effects in vitro, has attracted much attention for the treatment of anti-allergic diseases [13,16].

Menthone is rich in *Mentha* × *Piperita* L. essential oil and it has anti-inflammatory properties. It was found useful (possibly through percutaneous absorption) in treating *Schistosoma mansoni* infections, depressive disorders, gallbladder diseases, and DNA damage in sperm cells [17,18,19,20,21,22,23]. Moreover, menthone was found to alleviate type I allergic reactions via inhibiting histamine release from antigen-induced rat peritoneal mast cells [16]. However, the anti-allergic effects of volatile menthone have not yet been used to treat allergic asthma, in vivo. We hypothesized that menthone inhalation may provide anti-inflammatory and anti-allergic effects in patients with allergic asthma. To verify this, menthone was administered to treat ovalbumin (OVA)-sensitized and challenged BALB/c mice via inhalation. We measured the allergic inflammation mediator changes in the lungs and airways, sera, splenocytes, and peritoneal macrophages of the experiment mice. Relative expression amounts of six receptor genes related to allergic inflammation of the lungs and airways, including *epidermal growth factor receptor* (*Egfr*), *histamine receptor H1* (*Hrh1*), *tumor necrosis factor receptor superfamily, member 1a* (*Tnfrsf1a*), high affinity receptor for IgE, *FcεRI* (*Fcεr1a*), *chemokine (C-C motif) receptor 3* (*Ccr3*), and *chemokine (C-X-C motif) receptor 1* (*Cxcr1*), were quantitated.

## 2. Results

### 2.1. Effects of Menthone Inhalation on Feed Intake, Growth, and Visceral Organ Weights of OVA-Sensitized and Challenged Asthmatic Mice

The experimental design and mouse body weights from the experiment period are shown in Figure 1. The mean body weights of experiment mice in different groups slightly increased during the experiment period. DC group had significantly (*p* < 0.05) lower mean body weights than those of NSC group, suggesting OVA sensitization and challenge resulted in stress in the experiment mice (Figure 1C). There were no significant differences in body weights between DC and menthone-treated groups at the same experiment point, but menthone-treated groups had slightly higher body weights than those of DC group, indicating that the menthone inhalation period effectively alleviated the stress in the asthmatic mice.

Moreover, the initial and final body weights, food and energy intakes, as well as feed and energy efficiencies of OVA-sensitized and challenged mice are given in Appendix A. We observe that there were no significant differences in the initial and final body weights, body weight gain, feed intake, energy intake, feed efficiency, and energy efficiency among groups. However, DC group had slightly, but not significantly (*p* > 0.05), lower feed and energy efficiencies than those of NSC group, exhibiting that OVA sensitization and challenge caused particular stress in the experiment mice. Importantly, feed and energy efficiencies in the menthone-treated and dexamethasone-treated (positive control) mice were slightly increased throughout the experiment period, suggesting that both menthone inhalation and dexamethasone treatments alleviated the stress due to OVA sensitization and challenge.

In addition, we found that OVA sensitization and challenge (DC group) significantly (*p* < 0.05) decreased liver and thymus weights compared to those of the NSC group (Appendix A). Treatments with dexamethasone (PC group) significantly (*p* < 0.05) increased the liver weight but decreased the thymus weight compared to those of the DC group, indicating that dexamethasone treatment might induce inflammation in the liver but inhibit inflammatory immune responses in the thymus. However, menthone inhalation treatment might also inhibit inflammatory immune responses in the thymus compared to that of the DC group. Taken together, our results suggest that menthone inhalation treatments have no adverse side effects but may alleviate stress resulting from OVA sensitization and challenge.

### 2.2. Effects of Menthone Inhalation on Cellularity, Inflammatory Mediators, Th1/Th2, and Pro-/Anti-Inflammatory Cytokine Levels in the BALF of OVA-Sensitized and Challenged Asthmatic Mice

Our results showed that OVA sensitization and challenge markedly influenced the cellularity in the BALF (DC versus NSC) of the experiment mice. The OVA sensitization and challenge (DC group) significantly (*p* < 0.05) increased total cell numbers (from 8.8 ± 8.0 to 49.2 ± 33.6 × 10^5^ cells/mouse) and the percentage of eosinophils (from 0.00 ± 0.00% to 49.8 ± 14.8%), but slightly decreased (*p* > 0.05) the percentage of monocytes/macrophages (from 56.6 ± 24.7% to 19.5 ± 7.9%) and lymphocytes (Table 1). Our results indicated that the OVA-sensitized and challenged asthmatic animal model led to allergic inflammatory reactions in the lungs and airways; the model could be applied to evaluate the effects of menthone inhalation. In mice that inhaled menthone, there was a significant (*p* < 0.05) decrease in the total number of cells (from 49.2 ± 33.6 to 8.1 ± 5.7 × 10^5^ cells/mouse) and eosinophilic infiltration into BALF (from 49.8 ± 14.8% to 24.2 ± 6.9%), resembling (or may even be better than) the dexamethasone effect (PC group) (Table 1). Menthone inhalation also slightly decreased (*p* > 0.05) the percentage of lymphocytes (from 30.7 ± 7.2% to 18.4 ± 6.6%). Our results suggest that treatments with menthone inhalation may alleviate allergic inflammatory reactions in the lungs and airways by decreasing eosinophil and lymphocyte infiltration into the lungs and airways.

Table 2 shows the inflammatory mediators’ nitric oxide (NO), protein, and eotaxin levels in the BALF of asthmatic mice. We found that OVA sensitization and challenge (DC group) significantly (*p* < 0.05) increased the protein level (from 142 ± 25 to 257 ± 53 μg/mL) in the BALF compared to that in the NSC group, indicating that the asthmatic mice might suffer allergic inflammation of the lungs and airways. Menthone inhalation by the experiment mice slightly (*p* > 0.05) but dose-dependently decreased the protein (from 257 ± 53 to 172 ± 31 μg/mL) and eotaxin levels (from 284 ± 83 to 192 ± 78 pg/mL) in the BALF, resembling the dexamethasone effect (PC group) (Table 2). Our results suggest that menthone inhalation by the asthmatic experiment mice may alleviate allergic inflammatory reactions in the lungs and airways by decreasing both protein and eotaxin levels in the BALF.

Moreover, the results showed that high dose menthone inhalation (MH group) slightly (*p* > 0.05) inhibited IL-5 production, but the Th1 (IL-2 and IFN-γ) cytokines and Th1 (IL-2 + IFN-γ)/Th2 (IL-4 + IL-5) cytokine secretion ratio in the BALF significantly (*p* < 0.05) increased by menthone inhalation treatments (Table 3). The results suggest that menthone inhalation may have an anti-allergic effects on asthmatic mice by regulating the Th1/Th2 immune balance toward the Th1 immune balance in allergic asthma. Although pro-inflammatory IL-6 and TNF-α cytokine levels in the BALF increased, the pro-(IL-6)/anti-inflammatory (IL-10) cytokine ratio did not significantly (*p* > 0.05) increased in the BALF, suggesting that menthone inhalation might not cause obvious inflammatory responses in the lungs and airways. Dexamethasone administration (PC group) decreased all Th1/Th2 and pro-/anti-inflammatory cytokine levels in the BALF of OVA-sensitized and challenged asthmatic mice that might further alleviate an inflammatory status; however, it could not reverse Th1/Th2 immune balance in vivo.

To evaluate the overall effects of menthone inhalation on the lungs of the experiment mice, the histology of the lungs stained with H&E (Figure 2 and Figure 3) and PAS dyes (Appendix A) was further studied. Histopathological findings of the lungs were found in the OVA-induced bronchiolitis mice. Normal architectures of bronchioles of the lungs in a NSC mouse (Figure 2A,B), and OVA-induced lungs shown as focal inflammatory cell infiltration, mainly neutrophils, lymphocytes, and (fairly) eosinophils, around perivascular and peribronchial spaces, were, respectively, graded as slight (2) in the DC group (Figure 2C,D) and minimal (1) in the PC group (Figure 2E,F). Furthermore, lungs shown as focal inflammatory cell infiltration, mainly neutrophils. lymphocytes, and (fairly) eosinophils, around perivascular and peribronchial spaces, were, respectively, graded as slight (2) in the ML group (Figure 3A,B), MM group (Figure 3C,D), and minimal (1) in the MH group (Figure 3E,F). Based on the pathology–individual score of the lungs in the OVA-induced peribronchiolitis mice, menthone inhalation dose-dependently decreased the inflammatory cell infiltration into perivascular and peribronchial spaces in the lungs that might have alleviated allergic inflammation of the lungs and airways of the experiment mice (Table 4).

### 2.3. Effects of Menthone Inhalation on Lung Mast Cell Degranulation of OVA-Sensitized and Challenged Asthmatic Mice

We observed that menthone inhalation treatments in vivo significantly (*p* < 0.05) decreased the release of β-hexosaminidase from lung mast cells, particularly low dose menthone inhalation (Figure 4). Our results suggest that low dose menthone inhalation may alleviate allergic inflammation of the lungs and airways by stabilizing lung mast cells and inhibiting mast cell degranulation (Figure 4). However, dexamethasone administration (PC group) could not decrease lung mast cell degranulation compared to that of the DC group.

### 2.4. Effects of Menthone Inhalation on Antibody Secretion Levels in the Sera of OVA-Sensitized and Challenged Asthmatic Mice

The results showed that OVA-sensitized and challenged treatment (DC group) significantly (*p* < 0.05) increased OVA-specific IgG1, IgG2a, and IgE compared to those of the NSC group (Table 5). Importantly, high dose menthone inhalation (MH group) significantly (*p* < 0.05) increased the serum IgG2a/IgG1 (Th1-/Th2-type antibody) titer ratio compared to that of DC groups, suggesting that high dose menthone inhalation might alleviate systemic allergic status through relatively increasing Th1-type antibody titers via isotype switching in vivo. Interestingly, dexamethasone treatment (PC group) also alleviated the allergic inflammation stress due to OVA sensitization and challenge in the experiment mice compared to that of the DC group by increasing serum IgG2a/IgE (Th1-/Th2-type antibody) titer ratio. Our results evidence that menthone and dexamethasone may be applied to treat allergic asthma.

Moreover, the results showed that OVA sensitization and challenge significantly (*p* < 0.05) increased serum non-specific IgE, IgG, and IgG/IgM antibody ratio, but decreased the IgG/IgE ratio compared to those of the NSC group, suggesting that both Th1-type and Th2-type antibodies markedly increased in the OVA-sensitized and challenged asthmatic mice (Appendix A). However, the increased folds of total serum IgE were much higher than other serum non-specific antibodies, indicating that the asthmatic mice might suffer allergic symptom during this asthmatic period. Although low dose menthone inhalation (ML group) increased serum IgE levels, high dose menthone inhalation (MH group) significantly (*p* < 0.05) increased the IgG/IgM ratio compared to that of the DC group, suggesting that high dose menthone inhalation might improve the Th1/Th2 immune balance in asthmatic mice. Dexamethasone administration (PC group) increased IgA levels, but it could not significantly (*p* > 0.05) influence the Th1-/Th2-type antibody ratio in vivo.

### 2.5. Effects of Menthone Inhalation on Th1/Th2 Cytokine Secretion Levels in Splenocyte Cultures from OVA-Sensitized and Challenged Mice

Our results indicated that most spontaneous cytokine secretion levels, except IL-10, were lower than the sensitivity (<15.6 pg/mL) of ELISA kits used in the experiment (Table 6). However, low dose menthone inhalation (ML group) significantly increased (*p* < 0.05) IL-10 secretion levels in the splenocyte cultures. IL-10 is a Th2 and anti-inflammatory cytokine in vivo. Our results suggest that low dose menthone inhalation (ML group) may alleviate systemic inflammation status in the allergic asthmatic mice. To measure specific antigen stimulation effects on splenocytes, OVA was further selected to treat the splenocytes in vitro. We found that both Th1 and Th2 cytokine (except IFN-γ) secretion levels in the splenocyte cultures in the presence of OVA in the DC group significantly (*p* < 0.05) increased as compared to those in the NSC group. Moreover, menthone inhalation significantly (*p* < 0.05) induced IL-10 (Th2) and IFN-γ (Th1) cytokine secretion levels in the splenocyte cultures compared to those in the DC group. Importantly, OVA sensitization and challenge (DC group) significantly (*p* < 0.05) induced a Th2-skewed immune balance in the allergic asthmatic mice compared to that in the NSC group, reflected with markedly decreasing Th1 (IL-2 + IFN-γ)/Th2 (IL-4 + IL-5 + IL-10) cytokine secretion ratios. Most importantly, menthone inhalation dose-dependently and significantly (*p* < 0.05) restored Th1 immune balance in the experiment mice compared to that in the DC group, via obviously increasing Th1 (IL-2 + IFN-γ)/Th2 (IL-4 + IL-5 + IL-10) cytokine secretion ratios. Interestingly, dexamethasone treatment (PC group) also significantly (*p* < 0.05) enhanced the Th1 immune balance compared to that in the DC group, via increasing Th1 (IL-2 + IFN-γ)/Th2 (IL-4 + IL-5 + IL-10) cytokine secretion ratios. Our results suggest that menthone inhalation and dexamethasone treatment may alleviate allergic symptoms in the asthmatic experiment mice via improving Th2-skewed immune balance in allergic asthmatic mice.

### 2.6. Effects of Menthone Inhalation on Pro- and Anti-Inflammatory Cytokine Secretion Levels in Peritoneal Macrophage Cultures from OVA-Sensitized and Challenged Mice

The results showed that menthone inhalation mostly increased pro-inflammatory (TNF-α, IL-6 and IL-1β) cytokine secretion levels and pro-/anti-inflammatory secretion ratios (TNF-α/IL-10, IL-6/IL-10) by peritoneal macrophages either treated without or with LPS in vitro as compared to those in the DC group (Table 7). Most pro-inflammatory cytokines are classified as Th1-type cytokines. Importantly, our results suggest that menthone inhalation may alleviate Th2-skewed immune balance in allergic asthmatic mice by enhancing Th1-type cytokine secretions, but it may cause slight inflammation in the experiment mice. Moreover, dexamethasone treatment had anti-allergic effects similar to menthone inhalation by increasing pro-/anti-inflammatory secretion ratios (TNF-α/IL-10, IL-6/IL-10), but decreasing IL-10 and IL-1β secretion levels by peritoneal macrophages as compared to the DC group.

### 2.7. Effects of Menthone Inhalation on Relative Expression Amounts of Six Receptor Genes Related to Allergic Inflammation in the Lung and Airways Using a Two-Step Real Time Quantitative Polymerase Chain Reaction (qPCR) Assay

The results showed that OVA sensitization and challenge mostly increased *Egfr*, *Hrh1*, *Tnfrsf1a*, *Fc**ε**r1a*, *Ccr3*, and *Cxcr1* gene expression amounts compared to those of the NSC group, suggesting that OVA sensitization and challenge indeed induced allergic inflammation of the lungs and airways by increasing the gene expression amounts of six selected receptors related to allergic inflammation (Table 8). In particular, *Hrh1*, *Fcεr1a*, *Ccr3*, and *Cxcr1* gene expression amounts increased. Importantly, menthone inhalation mainly inhibited gene expression amounts of selected receptors, except *Tnfrsf1a* (just slight increase), compared to those of the DC group, suggesting that menthone inhalation may alleviate allergic inflammation status in the lungs and airways (Table 8). Most importantly, OVA sensitization and challenge (DC group) significantly (*p* < 0.05) increased *Ccr3* gene expression amounts compared to that of the NSC group; however, menthone inhalation similar to dexamethasone treatment significantly (*p* < 0.05) inhibited *Ccr3* gene expression amounts compared to that of the DC group. Our results evidence that menthone inhalation and dexamethasone treatment may alleviate allergic inflammation of the lungs and airways by inhibiting *Ccr3* gene expression amounts, possibly decreasing the expression amounts on lung mast cells and airway smooth muscle, and the recruitment of eosinophils, basophiles, and neutrophils into the lungs.

## 3. Discussion

In this study, menthone inhalation for short, recurrent times, such as during aromatherapy, was used to treat allergic asthmatic mice. We found that there were no significant differences in the body weights between the DC- and menthone-treated groups during the experiment period, but menthone-treated groups had slightly higher body weights than those of the DC group (Figure 1). Moreover, feed and energy efficiencies in the menthone-treated and dexamethasone-treated (positive control) mice were slightly higher than those of other experiment mice throughout the experiment period (Appendix A). We hypothesized that both menthone inhalation and dexamethasone treatments might have alleviated the stress due to OVA sensitization and challenge that resulted in fever and the loss of appetite in mice. Importantly, our results exhibited that OVA sensitization and challenge (DC group) significantly (*p* < 0.05) decreased the liver and thymus weights compared to those of the NSC group (Appendix A), while treatments with dexamethasone (PC group) significantly (*p* < 0.05) increased the liver weight but decreased the thymus weight compared to those of the DC group, implying that dexamethasone treatment might induce inflammation in the liver but inhibit inflammatory immune responses in the thymus. Similarly, treatment with menthone inhalation might also inhibit inflammatory immune responses in the thymus compared to that of the DC group. Based on feed intake, growth, and visceral organ weights of OVA-sensitized and challenged asthmatic mice, our results suggest that menthone inhalation, for short, recurrent times, such as during aromatherapy, do not have apparent adverse side effects but may alleviate asthmatic stress resulting from OVA sensitization and challenge. The present study is the first report to evidence that menthone inhalation, such as aromatherapy, may be applied to asthma treatment in the future.

T cell cytokine levels, differential cells counts, inflammatory mediators, etc., in the BALF may reflect the local asthma attack status and the effectiveness of asthma management through either classical or alternative therapies [1]. This study shows that menthone inhalation by the asthmatic experiment mice alleviated allergic inflammatory reactions in the lungs and airways by lowering the infiltration of eosinophils and lymphocytes (Table 1), decreasing both protein and eotaxin levels in the BALF (Table 2), but increasing Th1 (IL-2 and IFN-γ) cytokines and the Th1 (IL-2 + IFN-γ)/Th2 (IL-4 + IL-5) cytokine secretion ratio (Table 3). Importantly, the pro-(IL-6)/anti-inflammatory (IL-10) cytokine ratio did not significantly (*p* > 0.05) increase in the BALF, showing that menthone inhalation did not cause obvious inflammatory responses in the lungs and airways (Table 3). Taken together, our report, based on changes in cellularity, inflammatory mediators, Th1/Th2, and pro-/anti-inflammatory cytokine levels in the BALF of OVA-challenged asthmatic mice, suggest that menthone inhalation by asthmatic subjects may alleviate the allergic and inflammatory reactions in their lungs and airways.

Mast cell degranulation in the lungs and airways may be activated with its specific allergen through IgE cross-linking on a mast cell surface or by various secretagogues [3,4], and subsequently induce different allergic symptoms due to the release of different mediators, including histamine, tryptase, and β-hexosaminidase [1,2,4,17]. Importantly, we evidenced that menthone inhalation in vivo significantly (*p* < 0.05) decreased the release of β-hexosaminidase from lung mast cells (Figure 4), further suggesting that menthone inhalation may alleviate allergic inflammation of the lungs and airways by stabilizing lung mast cells and inhibiting mast cell degranulation (Figure 4). Natural mast cell stabilizing agents, including flavonoids, coumarins, phenols, terpenoids, and amino acids, have been suggested as mast cell stabilizers to prevent allergic reactions [3]. *Phlomis umbrosa* root decreases mast cell-dependent allergic reactions and inflammatory cytokine secretion [9]. Eucalyptus oil has been found to decrease allergic reactions and suppress mast cell degranulation by downregulating IgE-FcεRI signaling [7]. In the present study, we found that the anti-mast cell degranulation effect of menthone inhalation in the experiment was much better than that of dexamethasone treatment in vivo (PC group). Menthone may be developed as an alternative therapy, such as aerosol or aroma therapy, for the management of allergic asthma [1].

To unravel the possible target mechanisms of menthone inhalation, relative gene expression levels of six known receptors, *Egfr*, *Hrh1*, *Tnfrsf1a*, *Fcεr1a*, *Ccr3*, and *Cxcr1* genes (Appendix A), related to allergic inflammation in the mouse lungs of OVA-sensitized and challenged asthmatic mice were measured. Our results revealed that OVA sensitization and challenge almost increased *Egfr*, *Hrh1*, *Tnfrsf1a*, *Fc**ε**r1a*, *Ccr3*, and *Cxcr1* gene expression amounts, particularly *Hrh1*, *Fcεr1a*, *Ccr3*, and *Cxcr1*, compared to those of the NSC group, indicating that OVA sensitization and challenge indeed induced allergic inflammation of the lungs and airways through specific receptors (Table 8). Menthone inhalation inhibited all selected receptor (except *Tnfrsf1a*) gene expression amounts compared to those of the DC group, indicating that menthone inhalation may alleviate the allergic inflammation status in the lungs and airways (Table 8). Most importantly, OVA sensitization and challenge (DC group) significantly (*p* < 0.05) enhanced *Ccr3* gene expression amounts compared to that of the NSC group; however, treatments with menthone inhalation significantly (*p* < 0.05) inhibited *Ccr3* gene expression amounts compared to that of the DC group, indicating that menthone inhalation may alleviate allergic inflammation of the lungs and airways by decreasing *Ccr3* gene expression amounts. Ccr3; that is, C-C motif chemokine receptor type 3 (CD193), highly expressed in eosinophils, basophils, Th1 cells, Th2 cells, and airway epithelial cells for ligand eotaxins, RANTES, monocyte chemotactic protein (MCP)-2, MCP-3, and MCP-4, plays a vital role in promoting allergen-induced airway inflammation and hyperresponsiveness by mostly regulating eosinophils and mast cells [24,25,26]; however, Ccr3 has weak specificity in regard to activating basophils to release histamine [27]. It is of importance to Ccr3, in recruiting eosinophils and CD4^+^ T cells to the upper airway mucosa, to induce allergic symptoms [25,28,29]. The downregulation of murine Ccr3 results in the inhibition of proliferation and enhances apoptosis in eosinophils [30]. Decreased *Ccr3* gene expression amounts in the lungs might result from lower expression amounts on lung mast cells and the airway smooth muscle, and less recruitment of eosinophils, basophiles, and neutrophils into the lungs. Moreover, the present study evidenced that decreased *Ccr3* gene expression amounts may alleviate allergic inflammation of the lungs and airways by decreasing focal inflammatory cell infiltration, particularly lymphocytes and eosinophils (Table 1, Figure 2 and Figure 3, Table 4). The downregulation of Ccr3 may be developed as a promising approach for the treatment of allergic asthma [30].

Asthma is a Th2-polarized chronic allergic disease with local (the lungs and airways) and systemic inflammation that may reflect in the serum and immune cells [31]. We found that high dose menthone inhalation (MH group) significantly (*p* < 0.05) increased the serum IgG2a/IgG1 (Th1-/Th2-type antibody) titer ratio compared to that of DC group (Table 5), indicating that high dose menthone inhalation alleviated a systemic allergic status by modulating the OVA-specific antibody profile toward Th1-type antibody titers via isotype switching in vivo. Moreover, high dose menthone inhalation (MH group) significantly (*p* < 0.05) increased the IgG/IgM ratio compared to that of the DC group (Appendix A), indicating that high dose menthone inhalation improved Th1/Th2 immune balance in asthmatic mice. Furthermore, the OVA sensitization and challenge (DC group) significantly (*p* < 0.05) induced a Th2-skewed immune balance in the allergic asthmatic mice, reflected with markedly lower Th1 (IL-2 + IFN-γ)/Th2 (IL-4 + IL-5 + IL-10) cytokine secretion ratios (Table 6). It was of great importance that menthone inhalation appropriately restored Th1 immune balance in the experiment mice by increasing Th1 (IL-2 + IFN-γ)/Th2 (IL-4 + IL-5 + IL-10) cytokine secretion ratios in a dose-dependent manner. Our results evidence that menthone inhalation may alleviate Th2-skewed immune balance in the allergic asthmatic mice by enhancing M1 and Th1-type cytokine secretions by peritoneal macrophages (Table 7). Taken together, our results suggest that menthone inhalation may prevent and improve allergic inflammation in asthma patients by alleviating systemic and local Th2-skewed immune balance [31].

There are some limitations in the present study, even though some achievements were attained. This animal study primarily focused on Th1/Th2 cell responses in asthma; however, Th17 and Treg cells may play a part in severe asthma and should be explored in a future study [2]. In addition, a decrease in *ccr3* gene expression in the lung by menthone inhalation might not have been an entire mechanism but partially resulted from reduced infiltration of eosinophils, neutrophils, etc., into the lungs, due to less allergic inflammation in asthma cases. Our work is incomplete because the relevance of altered mRNA expression was not verified at the functional level. Moreover, the data will be more informative if gene expression amounts were analyzed in lavageable cells in addition to (or instead of) total lung RNA. Peritoneal macrophages were selected to assess the pro- and anti-inflammatory cytokine profiles in the present study. While this experiment is relevant, characterizing the cytokine profiles in BAL macrophages might be more appropriate. Total targets of menthone inhalation effects should also be studied further.

## 4. Materials and Methods

### 4.1. Materials

Menthone (C_10_H_18_O), a mono-terpenoid compound, was purchased at the highest available purity (>95%) from Sigma Co. (Sigma, 63680, St. Louis, MO, USA) and freshly dissolved in phosphate buffered saline (PBS) when used.

### 4.2. Experimental Design and Animals Grouping

BALB/cByJNarl female mice (7 weeks old) were provided by the National Laboratory Animal Center, National Applied Research Laboratories, Taipei, Taiwan. All experiment mice were housed in an animal room with 12-h dark/light cycles, 23 ± 2 °C and 50–75% relative humidity. Mice were fed a chow diet (laboratory standard diet, Diet MF 18, Oriental Yeast Co., Ltd., Osaka, Japan) and water ad libitum for one week before feeding on the AIN-76 feed to adapt to the environment. After this adaptation period, the experiment mice were randomly distributed into six groups (*n* = 15) at day -1, including non-sensitized control (NSC, normal mice), deke control (DC, a non-treatment control), menthone low dose inhalation (ML, 0.3 mg menthone/L inhalation at 2.8 mL/min for 25 min after 10 mg OVA/l challenge at 2.8 mL/min for 25 min, twice a day at day 31, 33, and 35), menthone medium dose inhalation (MM, 1.2 mg menthone/L inhalation at 2.8 mL/min for 25 min after 10 mg OVA/l challenge at 2.8 mL/min for 25 min, twice a day at day 31, 33, and 35), menthone high dose inhalation (MH, 1.2 mg menthone/L inhalation at 2.8 mL/min for 25 min after 10 mg OVA/l challenge at 2.8 mL/min for 25 min, twice a day at day 31, 33, and 35), and positive control (PC, 3 mg dexamethasone (DEX)/kg b.w. by gavage 2 h before the OVA challenge at day 31, 33, and 35) (Figure 1). NSC, DC, and PC groups received PBS instead of menthone inhalation. Three mice in the same group were housed in a stainless steel cage and individually earmarked. About 100 μL of serum sample was collected using a retro-orbital venous plexus puncture from each individual experiment mouse under anaesthetization with 2% isoflurane (2-chloro-2-(difluoromethoxy)-1,1,1-trifluoro-ethane, cat. no., 4900-1605, Panion & BF Biotech Inc., Taipei, Taiwan) with a vaporizer (CAS-01, Northern Vaporizer Limited, Cheshire, England, UK) at days −1, 7, and 21 to measure changes in serum antibody titers. The animal use protocol in this experiment was approved (IACUC No. 100-92) and performed according to the guidelines and regulations of the Institutional Animal Care and Use Committee (IACUC), National Chung Hsing University, Taiwan. Animal study results were recorded and reported in accord with the Animal Research guidelines: Reporting In Vivo Experiments (ARRIVE).

### 4.3. Establishment of OVA-Sensitized and Challenged Allergic Asthmatic Inflammation Mouse Model

The OVA-sensitized and challenged allergic asthmatic inflammation mouse model was established using OVA sensitization injected intraperitoneally (i.*p*.) and challenged with aerosolized-OVA to induce stronger airway inflammation [32,33,34,35,36,37]. Briefly, the experiment mice in all groups, except the NSC group, were sensitized *i*.*p*. with aliquots of 0.2 mL alum-precipitated antigen consisting of 8 μg OVA and 2 mg Al(OH)_3_ (Sigma-Aldrich Co., A8222, St. Louis, MO, USA) to prime the immune response at day 0. Two booster injections were, respectively, given at days 14 and 28. The NSC group just received alum-phosphate-buffered saline (PBS). Furthermore, OVA-sensitized experiment mice were, respectively, challenged at days 31, 33, and 35 with aerosolized OVA at a fixed concentration of 10 mg OVA/mL PBS and nebulization rate of 2.8 mL/min for 25 min twice a day using an ultrasonic nebulizer (sw918, Shinmed, Taipei, Taiwan). Each individual mouse was transiently held in a plastic murine restrainer under a big plastic chamber to evenly inhale the aerosolized reagent. The NSC group received aerosolized PBS only. At day 36, the experiment mice were bled with a retro-orbital venous plexus puncture under anesthesia with 2% isoflurane and immediately humanly killed using CO_2_ inhalation. Visceral tissues and organs were carefully cut, weighed, and analyzed. To evaluate the overall effects of menthone inhalation on the lungs of the experiment mice, the histology of the lungs was further studied.

### 4.4. Isolation of Bronchoalveolar Lavage Fluid (BALF) and Immune Cells, Peritoneal Macrophages, Splenocytes, as Well as Lung Mat Cells for Analyses

#### 4.4.1. Bronchoalveolar Lavage Fluid (BALF) Collection and Immune Cell Isolation for Differential Cell Count and Analysis

The experiment mice, after fasting for 12 h, were sacrificed. To harvest BALF, the lungs and airways of the experiment mice were aseptically lavaged using a cannula through the trachea with 5 aliquots of 0.6 mL sterile 0.85% sodium chloride (NaCl) solution. The BALF (ca. 3 mL) was collected and centrifuged at 4 °C, 400× *g* for 10 min. The supernatant of the BALF was collected for the inflammatory mediator assay, including nitric oxide, protein, and eotaxin levels. The measurements of the inflammatory mediators in the BALF were analyzed and performed as described previously [34]. The remaining cell pellet was harvested and resuspended in 1 mL of minimum essential medium (MEM medium) for the differential cell count. The differential cell count, including eosinophils, monocytes/macrophages, and lymphocytes, were stained, identified, and counted according to standard morphologic criteria using microscopy [34].

#### 4.4.2. Isolation of Primary Peritoneal Macrophages for Cultures

Immediately after BALF was harvested, the peritoneal macrophages of the experiment mice were isolated and performed as previously described [35,36,38,39]. Briefly, the isolated peritoneal macrophages (>95%) from each experiment mouse were adjusted to the cell density of 2 × 10^6^ cells/mL TCM medium for culture. The primary peritoneal macrophages (2 × 10^6^ cells/mL TCM medium, 0.50 mL/well) were, respectively, cultured with TCM medium (0.50 mL/well) and endotoxin lipopolysaccharide (LPS, 5.0 μg/mL, 0.50 mL/well, Sigma, L2654, St. Louis, MO, USA) in a 48-well plate for 48 h. Supernatants of cell cultures were harvested and stored at −80 °C for following pro-(IL-1β, IL-6, and TNF-α) and anti-inflammatory (IL-10) cytokine assays using a sandwich enzyme-linked immuno-absorbent assay (ELISA).

#### 4.4.3. Isolation of Primary Splenocytes for Cultures

After peritoneal macrophages were harvested, the spleen of each individual experiment mouse was removed aseptically to prepare single splenocytes [39,40,41]. The splenocytes suspension was adjusted to a fixed density of 1 × 10^7^ cells/mL in TCM medium for following culture experiments.

To measure the proliferation of splenocytes, the splenocytes (1 × 10^7^ cells/mL, 50 μL/well) were cultured in 96-well plates without or with specific mitogens, including OVA (60 μg/mL, 50 μL/well), LPS (a B-cell mitogen, 5.0 μg/mL, 50 μL/well), and concanavalin A (Con A, a T-cell mitogen, Sigma, L9132, St. Louis, MO, USA; 5.0 μg/mL, 50 μL/well), and the plate was incubated in a humidified incubator at 37 °C, 5% CO_2_, and 95% air for 72 h. The viable cells were determined using 3-(4,5-dimethylthiazol-2-yl)-2,5-diphenyltetrazolium bromide (MTT) assay [42,43]. The viable cells were presented as stimulation index (S.I.) compared to the mean absorbency (A) of the NSC group at 550 nm. Changes in S.I. in each individual experiment mouse among groups were computed using the following equation: S.I. = (A_treatment_ − A_mean of NSC group_)/(A_mean of NSC group_ − A_blank_).

To measure changes in cytokine secretions by splenocytes, the splenocytes (1 × 10^7^ cells/mL, 0.5 mL/well) were cultured in 48-well plates in the absence or presence of specific mitogen OVA (60 μg/mL, 0.5 mL/well) for 48 h [39]. The culture supernatant was harvested for measuring Th1 (IL-2 and IFN-γ)/Th2 (IL-4, IL-5 and IL-10) cytokine levels using ELISA.

#### 4.4.4. Isolation of Primary Lung Mast Cells for the Degranulation Assay

The lungs of the experiment mice were cut for lung mast cell isolation as previously described by Xie et al. [44] and Maximiano et al. [45]. The lungs were aseptically cut, put in a 3 cm cell-culture dish, and the blood was washed off with 3 mL of sterile 0.85% NaCl solution. Half of the lung tissue was cut and stored in liquid nitrogen for further total RNA extraction. The remaining lung half was finely cut using sterile scissors and the lung cells were extracted from the lung tissue [44,45]. The lung cell pellet was directly subjected to prepare lung mast cells with gravity sedimentation [46]. An aliquot of 3 mL of Histopaque-1119 (density = 1.119 g/mL, Sigma, 11191, St. Louis, MO, USA) was added to the lung cell pellet in a tube to resuspend the cells. Then, 3 mL of Histopaque-1077 (density = 1.077 g/mL, Sigma, 11171, St. Louis, MO, USA) and 3 mL of MEM-Alpha medium were sequentially added into the 15 mL tube. The tube was centrifuged for 10 min at 25 °C and 400× *g* to redistribute the cells. Lung mast cells were dispersed at the upper layer of Histopaque-1119 (density = 1.119 g/mL). An aliquot of 4.5 mL top supernatant was carefully pipetted to discard. An aliquot of 3.5 mL of lung mast cells in the tube was carefully pipetted and transferred to another 15 mL tube. An aliquot of 6.5 mL of HBSS buffer (free of calcium and magnesium ion) was added into the tube, mixed, and centrifuged at 25 °C and 400× *g* for 7 min. The supernatant was discarded. The lung mast cell pellet was harvested and washed with HBSS buffer twice to clear any Histopaque residue. The single lung mast cells were redispersed in MEM-Alpha medium. Twenty μL of lung mast cells were added to a 1.5 mL clean tube and stained with toluidine blue (a particular mast cell stain dye). The stained mast cells were computed with a hemocytometer. The purity of isolated mast cells was estimated higher than 95%. Finally, viable mast cells were calibrated to 1 × 10^6^ cells/mL MEM-Alpha medium with a hemocytometer using the trypan blue dye exclusion method.

To measure mast cell degranulation with β-hexosaminidase secretion assay, aliquots of 100 μL of lung mast cells (1 × 10^6^ cells/mL MEM-Alpha medium) from each individual experiment mouse were added into a 96-well plate to revive the cells at 37 °C for 10 min in a humidified incubator with 95% air and 5% CO_2_. The plate was centrifuged for 10 min at 400× *g* and 25 °C to remove the supernatant. Aliquots of 100 μL compound 48/80 (30 μg/mL) (Sigma, C2313, St. Louis, MO, USA) were pipetted into the wells to induce mast cell degranulation. The plate was further incubated at 37 °C for 1 h using a humidified incubator with 95% air and 5% CO_2_. The plate was then centrifuged at 4 °C and 400× *g* for 10 min. Aliquots of 100 μL of mast cell culture supernatants were, respectively, pipetted into 2 mL centrifuge tubes for the following β-hexosaminidase release assay. The mast cell pellets remaining in the wells were lysed with aliquots of 100 μL of 0.1% Triton X-100 solution (Sigma, 93443, St. Louis, MO, USA) for 30 min with gently shaking. The plate was centrifuged for 10 min at 4 °C and 400× *g*. Aliquots of 100 μL of the cell lysates were, respectively, harvested into 2 mL tubes for the following mast cell degranulation analysis with the β-hexosaminidase secretion assay [4,47,48].

Mast cell degranulation levels were determined by analyzing the release of β-hexosaminidase. Briefly, aliquots of 50 μL of the mast cell culture supernatants and 50 μL of the mast cell lysates from each individual mouse were, respectively, added into the wells in a 96-well plate. Aliquots of 50 μL substrate/well of 2 mM *p*-nitrophenyl-N-acetyl-β-D-glucosaminide (PNAG, Sigma, N9397, St. Louis, MO, USA), which dissolved in 100 mM, pH 4.5 citrate buffer (consisting of 100 mM sodium citrate and 98.7 mM HCl) were pipetted into the wells. The plate was further incubated at 37 °C for 1.5 h. Aliquots of 50 μL/well of 0.37 M, pH 10.7 carbonic acid buffer (0.26 M Na_2_CO_3_ and 0.11 M NaHCO_3_) were added to the wells to stop the enzyme reaction. The absorbance (A) at 405 nm of each well in the 96-well plate was determined with a microplate reader (Microplate Reader FLUOstar-Omega, 415-1103, Ortenberg, Germany). Mast cell degranulation was calculated in each corresponding biological determination and expressed as a percentage (%) using the following equation, mast cell degranulation (%) = [(A_supernatant_)/(A_supernatant_ + A_cell lysate_)] × 100.

### 4.5. Effects of Menthone Inhalation on Relative Expression Amounts of Six Receptor Genes Related to Allergic Inflammation in the Lungs and Airways Using a Two-Step Real Time Quantitative Polymerase Chain Reaction (qPCR) Assay

Half of the lung tissue of each individual mouse was cut and stored in liquid nitrogen for further total RNA extraction to evaluate the effects of menthone inhalation on the relative expression amounts of six particular receptor genes related to allergic inflammation in the lungs and airways (Appendix A). Briefly, the lung tissues from the experiment mice were, respectively, cut for total RNA extraction, synthesis of first-strand complementary DNA (cDNA) via reverse transcription, and determination of the relative expression folds of target genes in the lung samples using a real-time qPCR assay. Relative mRNA expression folds of six target genes in the lungs and airways, including *epidermal growth factor receptor* (*Egfr*), *histamine receptor H1* (*Hrh1*), *tumor necrosis factor receptor superfamily, member 1a* (*Tnfrsf1a*), high affinity receptor for IgE, *FcεRI* (*Fcεr1a*), *chemokine (C-C motif) receptor 3* (*Ccr3*), and *chemokine (C-X-C motif) receptor 1* (*Cxcr1*), were measured. The two-step real-time qPCR assay was manipulated as previously described [49,50]. The amount of mRNA expression was normalized to β-actin (a stable house-keeping gene) RNA in each individual sample. The primer target gene sequences are shown in Table 9. The relative expression of each mRNA species was measured with the comparative threshold cycle number (Ct) of the expression amount of target gene detected with fluorescence intensity using a real time qPCR analyzer [50,51]. Real time qPCR reactions were manipulated and quantitated with a real time rotary analyzer (Corbett Research Rotor-Gene 6000, Corbett Life Science, Sydney, Australia) by the following program: 95 °C, 15 min for hot-start activation and then 40 cycles of 95 ℃, 3 s for denaturation, 63 °C, 20 s for annealing, 72 °C, 1s for extension. Relative mRNA expression amounts among groups were computed and presented as the fold-change value (R) using the following equation: R = 2^−ΔΔCt^ [50,51]. ΔCt = (Ct_, target gene_ − Ct_, reference gene_); ΔΔCt = ΔCt (treatment X) − ΔCt (control) = (Ct_, target gene_ − Ct_, reference gene_)_treatment x_ − (C_t, target gene_ − Ct_, reference gene_)_control_.

### 4.6. Visceral Tissues Collection for Analyses

Visceral tissues, including the liver, kidney, thymus, epididymal fat, spleen, and lungs, of the experiment mice were cut and weighed. Absolute (ATW, gram weight) and relative tissue weights (RTW, % of body weight) of the organs compared to the body weight of each corresponding individual experiment mouse were analyzed [41].

### 4.7. Serum Non-Specific Antibody Quantification by ELISA

Serum total IgE, IgA, IgM, and IgG antibody levels were measured using the mouse IgE, IgA, IgM, and IgG ELISA quantitation kit (catalog number: E90-103, E90-115, E90-131, and E90-101, Bethyl Laboratories, Inc., Montgomery, TX, USA). After serum samples were appropriately diluted, serum total IgE, IgA, IgM, and IgG levels in the experiment mice were measured using a sandwich ELISA and performed as per the manufacturer’s instruction [33].

### 4.8. Serum OVA-Specific IgG1, IgG2a, and IgE Assay

Serum OVA-specific IgG1, IgG2a, and IgE titers of the experiment mice were determined as described previously [33]. Absorbency (A) at 450 nm was measured with the ELISA reader and converted into ELISA units (EU); EU = (A_sample_ − A_blank_)/(A_positive_ − A_blank_).

### 4.9. Measurement of Cytokine Levels in the BALF, Splenocytes, and Macrophages Cultures Using ELISA

Levels of Th1 (IL-2 and IFN-γ)/Th2 (IL-4 and IL-5) cytokines in the BALF, levels of Th1 (IL-2 and IFN-γ)/Th2 (IL-4, IL-5 and IL-10) cytokines secreted by the splenocytes, as well as levels of pro-(IL-1β, IL-6 and TNF-α)/anti-inflammatory (IL-10) cytokines secreted by the peritoneal macrophages from the experiment mice were measured using sandwich ELISA kits (mouse DuoSet ELISA Development system, R & D Systems Inc., Minneapolis, MN, USA). The protocol was performed according to the manufacture’s instruction [34,41]. The limit of detection (LOD) of the ELISA kits was <15.6 pg/mL.

### 4.10. Statistical Analysis

Results are presented as mean ± SD. Data were analyzed with one-way analysis of variance (ANOVA) followed by Duncan’s multiple range test. *p* < 0.05 refers to the statistically significant differences among groups. Statistical analyses were performed with SPSS version 12.0.

## 5. Conclusions

In the present study, menthone inhalation increased serum OVA-specific IgG2a/IgG1 and IgG2a/IgE ratios, increased Th1 cytokine production in the BALF of OVA-sensitized and challenged asthmatic mice, and decreased levels of nitric oxide, protein, and eotaxin. Moreover, menthone inhalation by asthmatic mice decreased eosinophilia and mast cell degranulation in their lungs and airways, but increased Th1 cytokine secretion ratios by splenocytes. Treatments with menthone inhalation significantly inhibited *Ccr3* gene expression amounts compared to that of the DC group in the lungs, suggesting that menthone inhalation may alleviate allergic inflammation of the lungs and airways by decreasing focal inflammatory cell infiltration. Taken together, our results evidence that menthone inhalation may improve airway allergic inflammation, mucin overproduction, eosinophils infiltration, Th1/Th2 immune balance, and relative expression levels of particular receptor genes related to allergic inflammation of the lungs and airways in the allergic asthmatic mice by regulating local and systemic immune responses in allergic asthma.

## Figures and Tables

**Figure 1 ijms-23-04011-f001:**
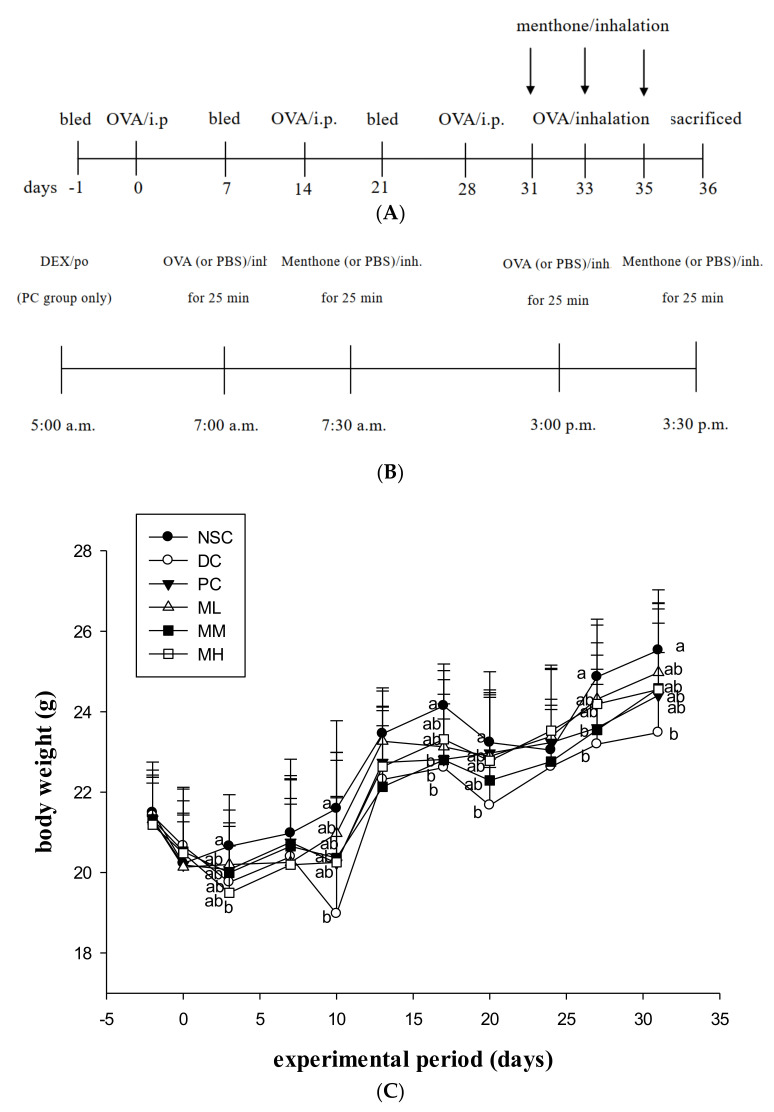
Effects of menthone inhalation with different doses on body weight changes of OVA/Al-sensitized and challenged BALB/c asthmatic mice. Values are means ± SD (*n* = 15). Values among DC, PC, ML, MM, and MH groups at the same experiment point not sharing a common small letter (a, b) are significantly different (*p* < 0.05) from each other, assayed by one-way ANOVA, followed by Duncan’s multiple range test. NSC: non-sensitized control; DC: dietary control; PC: positive control; ML: low dose menthone (0.3 mg/L); MM: medium dose menthone (1.2 mg/L); MH: high dose menthone (6 mg/L). (**A**) experimental design; (**B**) menthone inhalation treatment; (**C**) body weight changes of experiment mice.

**Figure 2 ijms-23-04011-f002:**
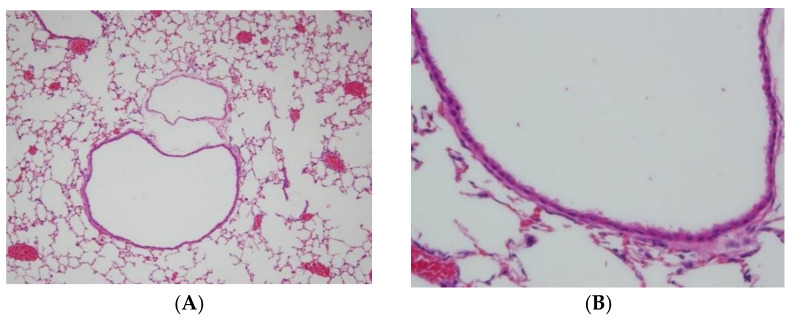
Histopathological findings of the lungs were found in the OVA-induced bronchiolitis mice. There were normal architectures of bronchioles of the lungs in a NSC mouse ((**A**,**B**), NCL3-2); OVA-induced lungs, shown as focal inflammatory cell infiltration, mainly neutrophils, lymphocytes, and (fairly) eosinophils, around perivascular and peribronchial spaces, which were, respectively, graded as slight (2) in the DC group ((**C**,**D**) DC3-1); and minimal (1) in the PC group ((**E**,**F**), PC3-3); H&E stain; 100 and 400×.

**Figure 3 ijms-23-04011-f003:**
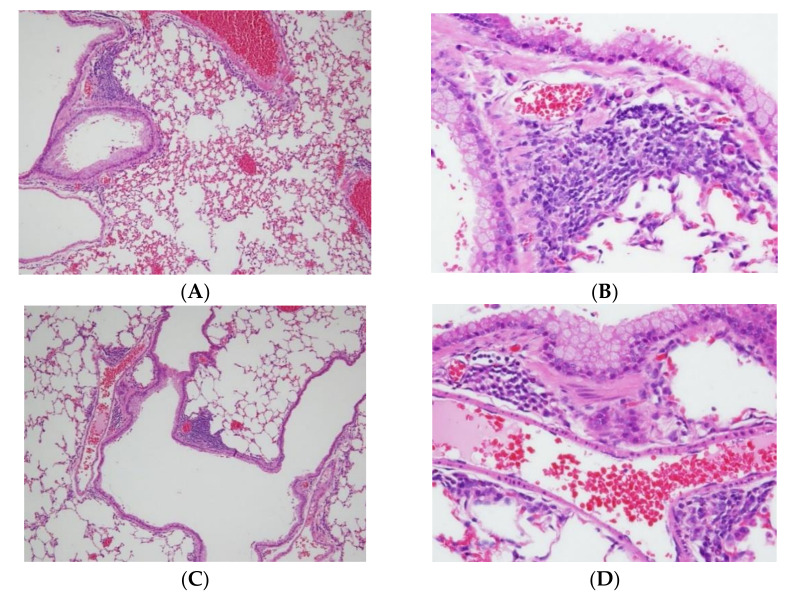
Histopathological findings of lungs were found in the OVA-induced bronchiolitis in mice. Lungs are shown as focal inflammatory cell infiltration, mainly neutrophils, lymphocytes, and (fairly) eosinophils, around perivascular and peribronchial spaces, which were, respectively, graded as slight (2) in the ML group ((**A**,**B**), ML3-1), MM group ((**C**,**D**), MM5-1), and minimal (1) in the MH group ((**E**,**F**), MH3-1), H&E stain; 100 and 400×.

**Figure 4 ijms-23-04011-f004:**
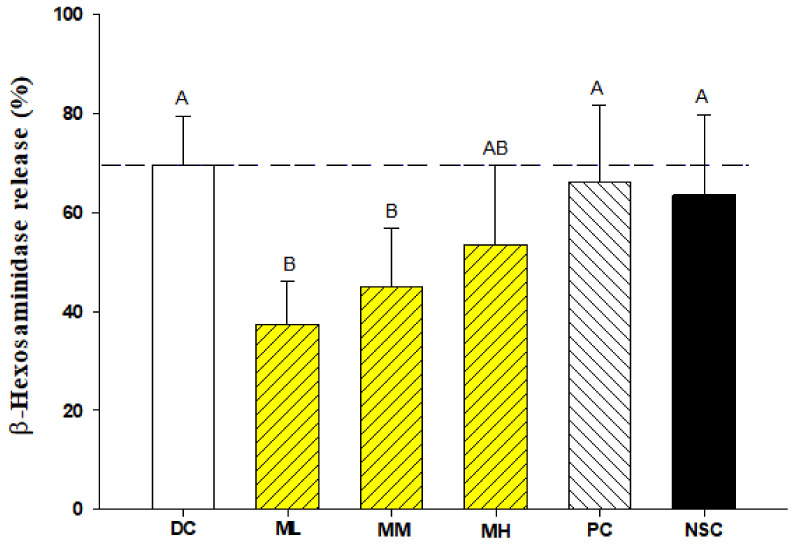
Effects of menthone inhalation with different doses on degranulation of lung mast cells of OVA/Al-sensitized and challenged BALB/c asthmatic mice. Values are means ± SD (*n* = 6). Bars not sharing a common capital letter (A, B) are significantly different (*p* < 0.05) from each other, assayed by one-way ANOVA, followed by Duncan’s multiple range test. The mouse lung mast cells were sensitized with compound 48/80 (30 μg/mL). NSC: non-sensitized control; DC: dietary control; PC: positive control; ML: low dose menthone (0.3 mg/L); MM: medium dose menthone (1.2 mg/L); MH: high dose menthone (6 mg/L).

**Table 1 ijms-23-04011-t001:** Effects of menthone inhalation with different doses on total cell numbers and differential cell distributions in the BALF of OVA/Al-sensitized and challenged BALB/c asthmatic mice.

Groups	Total Cell Number (×10^5^/Mouse)	Cell Distribution (%)		
		Eosinophils	Monocytes/Macrophages	Lymphocytes
DC	49.2 ± 33.6 ^A^	49.8 ± 14.8 ^A^	19.5 ± 7.9	30.7 ± 7.2 ^A,B^
ML	8.1 ± 6.3 ^B^	28.7 ± 8.9 ^B^	46.6 ± 6.6	24.7 ± 6.6 ^A,B^
MM	9.4 ± 6.7 ^B^	32.0 ± 12.9 ^B^	49.6 ± 12.4	18.4 ± 6.6 ^B^
MH	8.1 ± 5.7 ^B^	24.2 ± 6.9 ^B^	55.5 ± 5.2	20.4 ± 3.3 ^AB^
PC	22.7 ± 5.4 ^B^	21.4 ± 13.2 ^B^	43.6 ± 14.9	35.0 ± 22.7 ^A,B^
NSC	8.8 ± 8.0 ^B^	0.00 ± 0.00 ^C^	56.6 ± 24.7	43.5 ± 24.7 ^A^

Values are means ± SD (*n* = 6). Values within the same column not sharing a common capital letter (A, B, C) are significantly different (*p* < 0.05) from each other, assayed by one-way ANOVA, followed by Duncan’s multiple range test. NSC: non-sensitized control; DC: dietary control; PC: positive control; ML: low dose menthone (0.3 mg/L); MM: medium dose menthone (1.2 mg/L); MH: high dose menthone (6 mg/L).

**Table 2 ijms-23-04011-t002:** Effects of menthone inhalation with different doses on inflammatory mediator levels in the BALF of OVA/Al-sensitized and challenged BALB/c asthmatic mice.

Inflammatory Mediator Levels			
Groups	NO (μM)	Protein (μg/mL)	Eotaxin (pg/mL)
DC	3.24 ± 0.34 ^B,C^	257 ± 53 ^A^	284 ± 83 ^A,B^
ML	3.56 ± 0.89 ^A,B,C^	214 ± 46 ^A,B^	296 ± 91 ^A^
MM	3.39 ± 0.49 ^A,B,C^	212 ± 22 ^A,B^	198 ± 65 ^A,B^
MH	3.02 ± 0.74 ^C^	172 ± 31 ^B,C^	192 ± 78 ^B^
PC	4.15 ± 0.69 ^A^	231 ± 21 ^A^	166 ± 30 ^B^
NSC	3.99 ± 0.60 ^A,B^	142 ± 25 ^C^	179 ± 19 ^B^

Values are means ± SD (*n* = 5–6). Values within the same column not sharing a common capital letter (A, B, C) are significantly different (*p* < 0.05) from each other, assayed by one-way ANOVA, followed by Duncan’s multiple range test. NSC: non-sensitized control; DC: dietary control; PC: positive control; ML: low dose menthone (0.3 mg/L); MM: medium dose menthone (1.2 mg/L); MH: high dose menthone (6 mg/L).

**Table 3 ijms-23-04011-t003:** Effects of menthone inhalation with different doses on cytokine levels in the BALF of OVA/Al-sensitized and challenged BALB/c asthmatic mice.

Groups	Th1/Th2 Cytokine Levels in the BALF					
	IL-2 (pg/mL)	IFN-γ (pg/mL)	IL-4 (pg/mL)	IL-5 (pg/mL)	(IL-2 + IFN-γ)/(IL-4 + IL-5) Ratio (pg/pg)	
DC	51.1 ± 20.6 ^B^	3.10 ± 2.14 ^B^	66.2 ± 21.1 ^A,B^	152 ± 25 ^A,B^	0.25 ± 0.10 ^B^	
ML	83.1 ± 37.7 ^A^	5.96 ± 1.76 ^A^	85.7 ± 14.9 ^A^	205 ± 70 ^A^	0.32 ± 0.10 ^AB^	
MM	46.5 ± 17.4 ^B^	3.08 ± 1.61 ^B^	62.6 ± 26.6 ^A,B^	141 ± 55 ^A,B^	0.26 ± 0.07 ^B^	
MH	54.4 ± 24.3 ^A,B^	6.01 ± 1.90 ^A^	70.1 ± 30.5 ^A,B^	127 ± 57 ^B^	0.40 ± 0.05 ^A^	
PC	35.9 ± 9.4 ^B^	1.78 ± 0.66 ^B^	48.3 ± 4.3 ^B^	83 ± 19 ^B^	0.29 ± 0.06 ^A,B^	
NSC	53.0 ± 25.8 ^A,B^	3.38 ± 2.09 ^B^	53.9 ± 17.7 ^B^	96 ± 31 ^B^	0.33 ± 0.12 ^A,B^	
**Groups**	**Pro-/Anti-Inflammatory Cytokine Levels in the BALF**					
	**IL-1β (pg/mL)**	**IL-6 (pg/mL)**	**TNF-α (pg/mL)**	**IL-10 (pg/mL)**	**TNF-α/IL-10 (pg/pg)**	**IL-6/IL-10 (pg/pg)**
DC	23.2 ± 10.7 ^A,B^	30.0 ± 12.1 ^B,C^	14.7 ± 9.3 ^B^	56.6 ± 16.4 ^A,B^	0.26 ± 0.18	0.54 ± 0.15 ^A,B^
ML	34.4 ± 11.8 ^A^	61.7 ± 24.5 ^A^	34.8 ± 17.4 ^A^	74.2 ± 21.7 ^A^	0.47 ± 0.20	0.81 ± 0.10 ^A,B^
MM	34.1 ± 23.1 ^A^	36.8 ± 12.0 ^A,B,C^	11.9 ± 8.8 ^B^	52.8 ± 17.1 ^A,B^	0.21 ± 0.15	0.72 ± 0.21 ^A,B^
MH	24.3 ± 19.6 ^A,B^	52.5 ± 39.4 ^A,B^	30.0 ± 17.9 ^A^	55.3 ± 23.4 ^A,B^	0.40 ± 0.32	0.99 ± 0.83 ^A^
PC	6.3 ± 4.6 ^B^	14.0 ± 5.2 ^C^	6.2 ± 5.7 ^B^	36.5 ± 5.6 ^B^	0.22 ± 0.18	0.39 ± 0.16 ^B^
NSC	20.4 ± 17.2 ^A,B^	26.5 ± 9.0 ^B,C^	12.9 ± 9.1 ^B^	63.7 ± 40.0 ^A,B^	0.26 ± 0.17	0.56 ± 0.09 ^A,B^

Values are means ± SD (*n* = 5–6). Values within the same column not sharing a common capital letter (A, B, C) are significantly different (*p* < 0.05) from each other, assayed using one-way ANOVA, followed by Duncan’s multiple range test. The LOD of the kits used in this study was <15.6 pg/mL. “ND” means not detectable. NSC: non-sensitized control; DC: dietary control; PC: positive control; ML: low dose menthone (0.3 mg/L); MM: medium dose menthone (1.2 mg/L); MH: high dose menthone (6 mg/L).

**Table 4 ijms-23-04011-t004:** Pathology—individual score of the lungs in the OVA-induced peribronchiolitis mice.

Group/Organ	Histopathological Findings ^1^	Animal Code									
		NSC			DC				PC		
		m1	m2	m3	m1	m2	m3	m4	m1	m2	m3
Lung											
lesion score	Inflammatory cell infiltration, perivascular and peribronchial spaces	0	0	0	2	2	2	2	0	0	1
	Mean score		0			2				0.33	
**Group/Organ**	**Histopathological Findings ^1^**	**Animal Code**									
		**ML**			**MM**				**MH**		
		**m1**	**m2**	**m3**	**m1**	**m2**	**m3**		**m1**	**m2**	**m3**
Lung											
lesion score	Inflammatory cell infiltration, perivascular and peribronchial spaces	2	2	2	2	2	1		1	2	1
	Mean score		2			1.67				1.33	

^1^ The degree of the lesion was graded from one to five depending on severity: 0 = no significant lesion; 1 = minimal (<1%); 2 = slight (1–25%); 3 = moderate (26–50%); 4 = moderate/severe (51–75%); 5 = severe/high (76–100%).

**Table 5 ijms-23-04011-t005:** Effects of menthone inhalation with different doses of serum OVA-specific IgG1, IgG2a, and IgE antibody titers of OVA/Al-sensitized and challenged BALB/c asthmatic mice.

	OVA-Specific Antibody Levels (E.U.)			Th1/Th2 Antibody Ratio	
Groups	IgG1	IgG2a	IgE	IgG2a/IgG1	IgG2a/IgE
DC	1.46 ± 0.54 ^A^	1.07 ± 0.48 ^A,B^	1.33 ± 0.44 ^A,B^	0.82 ± 0.53 ^B^	0.79 ± 0.55 ^B^
ML	1.45 ± 0.40 ^A^	1.00 ± 0.52 ^B^	1.53 ± 0.43 ^A^	0.64 ± 0.38 ^B^	0.59 ± 0.37 ^B^
MM	1.47 ± 0.26 ^A^	1.43 ± 0.54 ^A,B^	1.28 ± 0.50 ^A,B^	1.01 ± 0.25 ^A,B^	0.97 ± 0.40 ^B^
MH	1.21 ± 0.31 ^A^	1.50 ± 0.76 ^A^	1.17 ± 0.29 ^B^	1.29 ± 0.56 ^A^	0.96 ± 0.42 ^B^
PC	1.45 ± 0.22 ^A^	1.41 ± 0.38 ^A,B^	1.20 ± 0.29 ^B^	0.99 ± 0.28 ^A,B^	1.37 ± 0.62 ^A^
NSC	0.00 ± 0.00 ^B^	0.00 ± 0.00 ^C^	0.00 ± 0.00 ^C^		

Values are means ± SD (*n* = 12–15). Values within the same column not sharing a common capital letter (A, B, C) are significantly different (*p* < 0.05) from each other, assayed by one-way ANOVA, followed by Duncan’s multiple range test. E. U. = (A_sample_ − A_blank_)/(A_positive_ − A_blank_); NSC: non-sensitized control; DC: dietary control; PC: positive control; ML: low dose menthone (0.3 mg/L); MM: medium dose menthone (1.2 mg/L); MH: high dose menthone (6 mg/L). Serum dilution: 1:500 for IgE detection, 1:5000 for IgG2a detection and 1:1,500,000 for IgG1 detection.

**Table 6 ijms-23-04011-t006:** Effects of menthone inhalation with different doses on Th1 and Th2 cytokine secretion profiles by splenocytes of OVA/Al-sensitized and challenged BALB/c asthmatic mice.

Cytokines	Groups	Treatment	
		Spon.	OVA
IL-4 (pg/mL)	DC	3.61 ± 2.28 ^B^	11.26 ± 3.41 ^A^
	ML	5.34 ± 2.05 ^A^	11.69 ± 5.35 ^A^
	MM	2.04 ± 1.54 ^C^	11.23 ± 3.61 ^A^
	MH	ND	6.56 ± 3.65 ^B^
	PC	4.17 ± 1.09 ^A,B^	5.63 ± 3.40 ^B^
	NSC	4.56 ± 1.83 ^A,B^	1.97 ± 1.44 ^C^
IL-5 (pg/mL)	DC	4.10 ± 2.91 ^A,B^	35.5 ± 25.7 ^B^
	ML	5.87 ± 5.21 ^A^	70.8 ± 40.4 ^A^
	MM	3.69 ± 3.39 ^A,B^	38.9 ± 28.3 ^B^
	MH	ND	38.4 ± 20.6 ^B^
	PC	2.34 ± 2.29 ^B,C^	1.8 ± 2.2 ^C^
	NSC	4.14 ± 3.25 ^A,B^	ND
IL-10 (pg/mL)	DC	26.4 ± 11.1 ^B^	214 ± 48 ^B^
	ML	67.0 ± 22.1 ^A^	299 ± 114 ^A^
	MM	18.2 ± 17.7 ^B^	306 ± 106 ^A^
	MH	7.3 ± 6.3 ^C^	281 ± 88 ^A^
	PC	6.3 ± 6.4 ^C^	69 ± 21 ^C^
	NSC	20.8 ± 11.4 ^B^	74 ± 11 ^C^
IL-2 (pg/mL)	DC	14.1 ± 4.4 ^A^	43.8 ± 5.4 ^A^
	ML	13.7 ± 3.2 ^A^	46.3 ± 7.3 ^A^
	MM	10.6 ± 2.7 ^A^	48.3 ± 12.7 ^A^
	MH	12.0 ± 5.2 ^A^	46.9 ± 9.6 ^A^
	PC	7.3 ± 5.0 ^B^	32.2 ± 11.0 ^B^
	NSC	12.6 ± 3.1 ^A^	13.6 ± 0.7 ^C^
IFN-γ (pg/mL)	DC	3.25 ± 3.53 ^B,C,D^	14 ± 5 ^D^
	ML	8.93 ± 4.78 ^A^	245 ± 170 ^A^
	MM	4.67 ± 2.93 ^B,C^	222 ± 107 ^A,B^
	MH	0.15 ± 0.31 ^D^	135 ± 95 ^B,C^
	PC	6.45 ± 4.75 ^A,B^	74 ± 45 ^C,D^
	NSC	2.67 ± 2.81 ^C,D^	205 ± 84 ^A,B^
(IL-2 + IFN-γ)/(IL-4 + IL-5 + IL-10) (pg/pg)	DC	0.42 ± 0.15 ^C^	0.20 ± 0.08 ^D^
	ML	0.34 ± 0.11 ^C^	0.45 ± 0.29 ^C,D^
	MM	0.51 ± 0.15 ^B,C^	0.83 ± 0.53 ^C,D^
	MH	0.55 ± 0.30 ^B,C^	1.22 ± 0.76 ^B,C^
	PC	1.23 ± 0.72 ^A^	2.00 ± 1.65 ^A,B^
	NSC	0.79 ± 0.51 ^B^	2.75 ± 1.43 ^A^

Values are means ± SD (*n* = 11–15). Values within the same column in the same item not sharing a common capital letter (A, B, C, D) are significantly different (*p* < 0.05) from each other, assayed by one-way ANOVA, followed by Duncan’s multiple range test. The LOD of the kits used in this study was <15.6 pg/mL. “ND” means not detectable. NSC: non-sensitized control; DC: dietary control; PC: positive control; ML: low dose menthone (0.3 mg/L); MM: medium dose menthone (1.2 mg/L); MH: high dose menthone (6 mg/L).

**Table 7 ijms-23-04011-t007:** Effects of menthone inhalation with different doses on pro- and anti-inflammatory cytokine secretion by peritoneal cells of OVA/Al-sensitized and challenged BALB/c asthmatic mice.

Cytokines Secreted by Splenocytes	Groups	Treatment	
		Spon.	LPS
TNF-α (pg/mL)	DC	20 ± 19 ^B^	724 ± 314 ^B,C^
	ML	674 ± 492 ^A^	1223 ± 562 ^A^
	MM	492 ± 354 ^A^	1220 ± 440 ^A^
	MH	32 ± 31 ^B^	944 ± 220 ^A,B^
	PC	ND	532 ± 282 ^C^
	NSC	53 ± 31 ^B^	933 ± 117 ^B^
IL-6 (pg/mL)	DC	39 ± 27 ^C^	1366 ± 568 ^C,D^
	ML	1208 ± 705 ^A^	2907 ± 832 ^A^
	MM	798 ± 615 ^B^	1593 ± 697 ^C^
	MH	56 ± 45 ^C^	2240 ± 828 ^B^
	PC	34 ± 28^C^	900 ± 249 ^D^
	NSC	91 ± 51 ^C^	1477 ± 406 ^C^
IL-10 (pg/mL)	DC	16.7 ± 9.0 ^A^	128 ± 28 ^B^
	ML	15.8 ± 6.1 ^A,B^	122 ± 62 ^B^
	MM	12.1 ± 7.1 ^A,B^	118 ± 58 ^B^
	MH	6.3 ± 4.5 ^C^	136 ± 66 ^B^
	PC	11.1 ± 3.6 ^B^	38 ± 22 ^C^
	NSC	13.6 ± 3.8 ^A,B^	276 ± 187 ^A^
IL-1β (pg/pg)	DC	0.99 ± 1.05 ^B,C^	22.8 ± 10.6 ^A^
	ML	6.52 ± 4.19 ^A^	28.3 ± 15.9 ^A^
	MM	2.67 ± 3.21 ^B^	29.4 ± 16.6 ^A^
	MH	ND	25.4 ± 14.3 ^A^
	PC	ND	ND
	NSC	0.18 ± 0.25 ^C^	29.6 ± 11.9 ^A^
TNF-α/IL-10 (pg/pg)	DC	1.6 ± 1.5 ^C^	3.8 ± 1.9 ^D^
	ML	43.4 ± 15.0 ^A^	9.9 ± 5.7 ^A,B^
	MM	25.3 ± 24.1 ^B^	8.4 ± 4.4 ^B,C^
	MH	2.1 ± 2.3 ^C^	6.1 ± 2.7 ^C,D^
	PC	ND	13.0 ± 4.6 ^A^
	NSC	3. 5 ± 1.8 ^C^	3.8 ± 2.1 ^D^
IL-6/IL-10 (pg/pg)	DC	1.9 ± 1.0 ^B^	6.9 ± 4.2 ^B^
	ML	57.0 ± 31.9 ^A^	24.4 ± 17.4 ^A^
	MM	30.7 ± 30.8 ^A^	19.9 ± 13.9 ^A^
	MH	6.3± 4.2 ^B^	19.3 ± 11.3 ^A^
	PC	2.9 ± 2.5 ^B^	24.1 ± 9. 7 ^A^
	NSC	8.1 ± 3.1 ^B^	7.3 ± 4.3 ^B^

Values are means ± SD (*n* = 11–15). Values within the same column in the same item not sharing a common capital letter (A, B, C, D) are significantly different (*p* < 0.05) from each other, assayed by one-way ANOVA, followed by Duncan’s multiple range test. The LOD of the kits used in this study was <15.6 pg/mL. “ND” means not detectable. NSC: non-sensitized control; DC: dietary control; PC: positive control; ML: low dose menthone (0.3 mg/L); MM: medium dose menthone (1.2 mg/L); MH: high dose menthone (6 mg/L).

**Table 8 ijms-23-04011-t008:** Effects of menthone inhalation with different doses on gene relative expression levels of receptors related to allergic inflammation in the mouse lungs of OVA/Al-sensitized and challenged BALB/c asthmatic mice.

Gene Name of Receptors	Relative Gene Expression (Fold)					
	Groups					
	DC	ML	MM	MH	PC	NSC
*Egfr*	1.00 ± 0.00	0.81 ± 0.4	0.42 ± 0.26	0.68 ± 0.15	0.85 ± 0.39	0.98 ± 0.52
*Hrh1*	1.00 ± 0.00	0.68 ± 0.55	0.37 ± 0.27	0.63 ± 0.24	0.58 ± 0.28	0.36 ± 0.10
*Tnfrsf1a*	1.00 ± 0.00	1.00 ± 0.70	1.30 ± 0.88	1.46 ± 0.91	1.33 ± 0.38	0.58 ± 0.30
*Fcεr1a*	1.00 ± 0.00 ^A,B^	0.36 ± 0.29 ^A,B^	0.16 ± 0.09 ^B^	0.47 ± 0.38 ^A,B^	0.65 ± 0.32 ^A^	0.42 ± 0.27 ^A,B^
*Ccr3*	1.00 ± 0.00 ^A^	0.24 ± 0.11 ^B^	0.20 ± 0.15 ^B^	0.26 ± 0.10 ^B^	0.32 ± 0.21 ^B^	0.41 ± 0.07 ^B^
*Cxcr1*	1.00 ± 0.00 ^A,B,C^	0.34 ± 0.17 ^C^	0.73 ± 0.59 ^B,C^	1.43 ± 0.68 ^A,B^	1.51 ± 0.35 ^A^	0.28 ± 0.05 ^C^

Values are means ± SD (*n* = 3). Values within same row not sharing a common capital letter (A, B, C) are significantly different (*p* < 0.05) from each other, assayed by one-way ANOVA, followed by Duncan’s multiple range test. The relative amounts among groups between target genes and a house-keeping gene β-actin are presented using the 2^-ΔΔCt^ equation, where ΔΔCt = (Ct target gene − Ct β-actin) _treatment x_ − (Ct target gene − Ct β-actin)_control._ NSC: non-sensitized control; DC: dietary control; PC: positive control; ML: low dose menthone (0.3 mg/L); MM: medium dose menthone (1.2 mg/L); MH: high dose menthone (6 mg/L). *Egfr*: epidermal growth factor receptor; *Hrh1*: histamine receptor H1; *Tnfrsf1a*: tumor necrosis factor receptor superfamily, member 1a; *Fcεr1a*: Fc receptor, IgE, high affinity I, alpha polypeptide; *Ccr3*: chemokine (C-C motif) receptor 3; *Cxcr1*: chemokine (C-X-C motif) receptor.

**Table 9 ijms-23-04011-t009:** Primer sequences used for detection of receptors related to allergic inflammation and a house-keeping gene β-actin using real time qPCR assays.

Gene Name ^b^	Primer Sequences	Length (bp) ^a^
*Egfr*	FW: GGGAGGAGGAGAGGAGAACTG	200
	RV: GTGGTGGGCAGGTGTCTTTGC	200
*Hrh1*	FW: CCTCATCTACCCGCTGTGCAAC	200
	RV: CTCAGCCCAGGACCTTCGATTC	200
*Tnfrsf1a*	FW: CACGAGGACACGCTGGAAGTAG	200
	RV: CCACAGGGAGTAGGGCATCTAG	200
*Fcεr1a*	FW: GTAACGCAAGATTGGCTGCTCCTTC	200
	RV: TGGTAGGTGCCACTGTCATTCAGTG	200
*Ccr3*	FW: TGGCACACAGACCCTAGAAATCTC	200
	RV: GTTGAGTCTCTGAACGCATCACAG	200
*Cxcr1*	FW: CCGATCCGTCATGGATGTCTACG	200
	RV: GGTATCGGTCCACACTGATGCAG	200
*β-actin*	FW: TGCCCATCTACGAGGGCTATGC	200
	RV: AGTGGCCATCTCCTGCTCGAAG	200

^a^ amplicon length. ^b^ *Egfr*: epidermal growth factor receptor; *Hrh1*: histamine receptor H1; *Tnfrsf1a*: tumor necrosis factor receptor superfamily, member 1a; *Fcεr1a*: Fc receptor, *IgE*, high affinity I, alpha polypeptide; *Ccr3*: chemokine (C-C motif) receptor 3; *Cxcr1*: chemokine (C-X-C motif) receptor.

## Data Availability

The authors declare that all data supporting this study are available within the manuscript or can be obtained from the authors upon request.

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
