# Peer review of "Menthone Inhalation Alleviates Local and Systemic Allergic Inflammation in Ovalbumin-Sensitized and Challenged Asthmatic Mice"

_ijms, 2022, doi:10.3390/ijms23074011_

Round 1

Reviewer 1 Report

The manuscript titled “Menthone Inhalation Alleviates Local and Systemic Allergic Inflammation in Ovalbumin-sensitized and Challenged Asthmatic Mice” presents effects of menthone inhalation on allergic inflammation. The manuscript is written well. There is minor typo and graphical error found for correction.

Minor:

  1. Please readjust the Figure 1 (A) and (B).
  2. Adjust alignment for all the tables in the manuscript. Some data points are moved within the columns. Please recheck it.
  3. Some variation on font style and size in table and graphs. Please make it uniform for publication purpose.

Author Response

Response to Reviewer 1 Comments

The manuscript titled “Menthone Inhalation Alleviates Local and Systemic Allergic Inflammation in Ovalbumin-sensitized and Challenged Asthmatic Mice” presents effects of menthone inhalation on allergic inflammation. The manuscript is written well. There is minor typo and graphical error found for correction.

Response: We thank the Reviewer’s comments. Responses to the Reviewer’s comments together with the itemized changes have been made.

Minor:

Please readjust the Figure 1 (A) and (B).

Adjust alignment for all the tables in the manuscript. Some data points are moved within the columns. Please recheck it.

Some variation on font style and size in table and graphs. Please make it uniform for publication purpose.

Response: We thank the Reviewer’s remind. We have carefully checked the original submitted manuscript. There is no problem in Figure 1. We think that there is error during the reproduction process by the editorial office. In addition, the font style and size in table and graphs in the revised manuscript have been made uniformly.

Reviewer 2 Report

Prior studies reported the anti-inflammation properties of menthone in various disease models (eg. Schistosomiasis, gallstone disease). Although one recent study (PMID: 33374928) using allergic asthma model reported menthone role in attenuation of IL6 signaling and amelioration of asthma, no further exploration was done using menthone in asthma. In this manuscript, Su et al., investigated the therapeutic benefits of menthone inhalation particularly on inflammation using a ova-induced allergic asthma model. In this study, authors have evaluated body weight, histopathological examination of lungs for inflammation, mast cell degranulation, serum levels of IgG2a/IgG1, splenocyte release of (Th1 and Th2) cytokine profile and macrophage inflammatory cytokine profile. Overall, this is a comprehensive work to determine the anti-inflammatory effects of menthone against ova challenge. However, there are some weaknesses in this study.

Major:

  1. In Table 2, authors reported that menthone inhalation significantly reduced the eosinophil recruitment. Although in Table 4, author’s data indicated relative Th1 and Th2 cytokine ration, authors did not establish whether menthone inhibits the production of cytokines responsible for eosinophilic infiltration (eg. IL5).
  2. For the histopathology figures shown, a low power entire lung section should be provided as supplement.
  3. In Figure 7, authors choose lung tissue for assessment of inflammation-associated receptor gene expression. This work seems incomplete because the relevance of altered mRNA expression was not verified at the functional level.
  4. In Figure 7, in addition to or instead of total lung RNA, if gene expression studies were done in lavageable cells, the data will be more informative.
  5. For Table 8, authors choose peritoneal macrophages to assess the pro- and anti-inflammatory cytokine profile. While this experiment is relevant, characterizing cytokine profile in BAL macrophages could be more appropriate.

Minor:

  1. Author’s attention is necessary in arranging the Figure 1 which at least in reviewer’s copy was fully misaligned!

Author Response

Response to Reviewer 2 Comments

Prior studies reported the anti-inflammation properties of menthone in various disease models (eg. Schistosomiasis, gallstone disease). Although one recent study (PMID: 33374928) using allergic asthma model reported menthone role in attenuation of IL6 signaling and amelioration of asthma, no further exploration was done using menthone in asthma. In this manuscript, Su et al., investigated the therapeutic benefits of menthone inhalation particularly on inflammation using a ova-induced allergic asthma model. In this study, authors have evaluated body weight, histopathological examination of lungs for inflammation, mast cell degranulation, serum levels of IgG2a/IgG1, splenocyte release of (Th1 and Th2) cytokine profile and macrophage inflammatory cytokine profile. Overall, this is a comprehensive work to determine the anti-inflammatory effects of menthone against ova challenge. However, there are some weaknesses in this study.

Response: We thank the Reviewer’s comments. Responses to the Reviewer’s comments together with the itemized changes have been made.

Major:

  • In Table 2, authors reported that menthone inhalation significantly reduced the eosinophil recruitment. Although in Table 4, author’s data indicated relative Th1 and Th2 cytokine ration, authors did not establish whether menthone inhibits the production of cytokines responsible for eosinophilic infiltration (eg. IL5).

Response: Page 8 lines 1-2 (revised manuscript): We thank the Reviewer’s comments. In fact, high dose menthone inhalation (MH group) slightly (P > 0.05) inhibited IL-5 production, but significantly increased (IL-2+IFN-γ)/(IL-4+IL-5) ratio (pg/pg) in BALF (Table 4). We think that it has evidenced a menthone effect on IL-5 production in vivo. The words “high dose menthone inhalation (MH group) slightly (P > 0.05) inhibited IL-5 production, but” have been added in the revised manuscript.

  • For the histopathology figures shown, a low power entire lung section should be provided as supplement.

Response: Pages 8, 54, 55 and 56 (revised manuscript): We thank the Reviewer’s suggestion. We did not have a low power entire lung section, however we can provide the lung section stained with PAS dye as supplement in the revised manuscript (Figures S1 and S2).

  • In Figure 7, authors choose lung tissue for assessment of inflammation-associated receptor gene expression. This work seems incomplete because the relevance of altered mRNA expression was not verified at the functional level. In Figure 7, in addition to or instead of total lung RNA, if gene expression studies were done in lavageable cells, the data will be more informative.

Response: Pages 18 lines 4-7 (revised manuscript): We agree the Reviewer’s comments. The description “Our work is still incomplete because the relevance of altered mRNA expression was not verified at the functional level. Besides, the data will be more informative if gene expression amounts were analyzed in lavageable cells in addition to or instead of total lung RNA.” has been added into the discussion section in the revised manuscript.

  • For Table 8, authors choose peritoneal macrophages to assess the pro- and anti-inflammatory cytokine profile. While this experiment is relevant, characterizing cytokine profile in BAL macrophages could be more appropriate.

Response: Pages 18 lines 7-10 (revised manuscript): We agree the Reviewer’s comments. The description “Peritoneal macrophages were selected to assess the pro- and anti-inflammatory cytokine profile in the present study. While this experiment is relevant, characterizing cytokine profile in BAL macrophages might be more appropriate.” has been added into the discussion section in the revised manuscript.

Minor:

Author’s attention is necessary in arranging the Figure 1 which at least in reviewer’s copy was fully misaligned!

Response: We thank the Reviewer’s remind. We have carefully checked the original submitted manuscript. There is no problem in Figure 1. We think that there is error during the reproduction process by the editorial office.